# Differential Small RNA Responses against Co-Infecting Insect-Specific Viruses in *Aedes albopictus* Mosquitoes

**DOI:** 10.3390/v12040468

**Published:** 2020-04-21

**Authors:** Lionel Frangeul, Hervé Blanc, Maria-Carla Saleh, Yasutsugu Suzuki

**Affiliations:** Institut Pasteur, Viruses and RNA Interference Unit, CNRS Unité Mixte de Recherche 3569, 75724 Paris CEDEX 15, France; lionel.frangeul@pasteur.fr (L.F.); herve.blanc@pasteur.fr (H.B.)

**Keywords:** *Aedes albopictus*, insect-specific viruses, co-infection, small interfering RNA, PIWI-interacting RNA, reproductive tissues, sex difference

## Abstract

The mosquito antiviral response has mainly been studied in the context of arthropod-borne virus (arbovirus) infection in female mosquitoes. However, in nature, both female and male mosquitoes are frequently infected with insect-specific viruses (ISVs). ISVs are capable of infecting the reproductive organs of both sexes and are primarily maintained by vertical transmission. Since the RNA interference (RNAi)-mediated antiviral response plays an important antiviral role in mosquitoes, ISVs constitute a relevant model to study sex-dependent antiviral responses. Using a naturally generated viral stock containing three distinct ISVs, *Aedes* flavivirus (AEFV), Menghai rhabdovirus (MERV), and Shinobi tetra virus (SHTV), we infected adult *Aedes albopictus* females and males and generated small RNA libraries from ovaries, testes, and the remainder of the body. Overall, both female and male mosquitoes showed unique small RNA profiles to each co-infecting ISV regardless of the sex or tissue tested. While all three ISVs generated virus-derived siRNAs, only MERV generated virus-derived piRNAs. We also studied the expression of PIWI genes in reproductive tissues and carcasses. In contrast to Piwi5-9, Piwi1-4 were abundantly expressed in ovaries and testes, suggesting that Piwi5-9 are involved in exogenous viral piRNA production. Together, our results show that ISV-infected *Aedes albopictus* produce viral small RNAs in a virus-specific manner and that male mosquitoes mount a similar small RNA-mediated antiviral response to that of females.

## 1. Introduction

Mosquitoes are efficient vectors for various human pathogens. *Aedes* and *Culex* are two major genera of mosquitoes that transmit medically important arthropod-borne viruses (arboviruses) such as dengue, West Nile and Zika viruses. Recent metagenomics studies have shed light on the virome of these mosquito vectors, revealing diverse virus communities largely composed of insect-specific viruses (ISVs) that specifically infect invertebrates but not vertebrates [1,2,3,4,5,6,7]. ISVs have been shown to negatively regulate some arbovirus infections in mosquito cell cultures and in vivo [8,9,10,11,12,13,14]. For instance, two ISVs, Menghai rhabdovirus (MERV) and Shinobi tetravirus (SHTV), which were identified in an *Aedes* (*Ae*.) *albopictus* C6/36 cell line from the Japanese Collection of Research Bioresources (JCRB), suppressed Zika virus replication in vitro [14]. Despite increased attention to ISVs, little is known about mosquito-ISV interactions such as the natural infection routes in vivo to the immune responses to ISVs. For example, although some ISVs induce cytopathic effects in mosquito cell cultures, few studies have investigated in vivo pathogenesis. It has been suggested that mosquitoes establish a tolerant state to ISVs as well as to arboviruses and that this viral tolerance is one of the critical factors influencing mosquito vector competence [15,16,17]. Therefore, understanding the mosquito antiviral response to ISVs could provide new insights into the viral tolerance mechanisms of mosquitoes.

The small interfering RNA (siRNA) pathway is considered to be a major antiviral defense mechanism in insects, including vector mosquitoes [18]. The cellular RNase-III enzyme Dicer-2 (DCR2) recognizes viral dsRNA forms, which are mostly viral replication intermediates and viral secondary structures, and processes them into 21 nt-long siRNAs. The siRNA duplexes are loaded into the RNA-induced silencing complex (RISC) containing Argonaute2 (Ago2), which cleaves the target viral RNA under the guide of siRNAs. Some mosquito viruses have evolved avoidance mechanisms to the siRNA pathway, demonstrating that virus-derived siRNAs (vsiRNAs) play an important antiviral function in mosquito vectors [19,20,21,22,23,24].

The P-element induced wimpy testis (PIWI)-interacting RNA (piRNA) pathway has been demonstrated to control transposable elements (TEs) in the germ line. The biogenesis of piRNAs is well characterized in *Drosophila melanogaster* and reviewed in [25]. piRNAs can be divided into two different classes: primary piRNAs and secondary piRNAs. Both types of piRNAs show a similar size distribution of 25–30 nt. Primary piRNAs are generated from single-stranded precursor transcripts derived from piRNA clusters enriched in transposon remnants, and their first nucleotide is typically a uridine, referred to as the 1U bias. Secondary piRNAs are produced by cleavage of complementary transcripts directed by primary piRNAs and often contain an adenine at the 10th nucleotide position, called the 10A bias. In addition to piRNA cluster-derived piRNAs, mosquitoes have been shown to produce viral piRNAs (vpiRNAs) from different replicating viruses such as flaviviruses, alphaviruses and rhabdoviruses [15,26,27,28,29,30,31,32]. This is in contrast with *Drosophila melanogaster*, where vpiRNAs are not produced during virus infections [33]. The process leading to the production of vpiRNAs remains unknown. Mosquitoes have an expanded PIWI family compared to *Drosophila*; *Ae. aegypti* has eight PIWI genes (Piwi1-7 and Ago3) and *Ae. albopictus* was estimated to have ten PIWI genes (Piwi1-9 and Ago3) [34,35]. This expansion could explain the production of vpiRNAs in mosquitoes. The antiviral potential of vpiRNAs remains unclear and the function of PIWI proteins in *Ae. albopictus* remains to be determined. Functional studies of PIWI proteins have been primarily conducted in the *Ae. aegypti* cell line Aag2 using arboviruses. Gene silencing of Piwi5 and Ago3 decreased vpiRNA biogenesis without impacting the replication of the Semliki Forest virus (SFV), Sindbis virus (SINV) or dengue virus [26,27,29,31,36]. Further experiments suggested that Piwi4 was antiviral against SFV, but not through the piRNA pathway [27]. More recently, it has been suggested that Piwi4 is involved in the maturation of both siRNAs and piRNAs [37]. 

Unlike arboviruses, ISVs are likely to be commensal in mosquito hosts and are considered to be maintained in nature by vertical transmission [38,39,40,41,42,43]. Due to their mode of transmission, ISVs provide a great opportunity to better understand the mosquito antiviral response in females as well as in males, especially antiviral responses such as the piRNA pathway in reproductive organs and the germ line. To date, studies on ISV-derived small RNA production have been limited to mosquito cell lines [28,30]. Here, we investigated the small RNA responses to ISVs in individual mosquitoes. A previous study identified *Aedes* flavivirus (AEFV) in a field-collected *Ae. albopictus* individual [44]. The AEFV strain was isolated in the C6/36 cell line from JCRB, which is persistently infected with two other ISVs, MERV and SHTV [44]. Consequently, AEFV stocks generated in these cells contain three ISVs that are phylogenetically distant: AEFV belongs to the *Flaviviridae* family and is a positive-sense single-stranded RNA (ssRNA) virus, MERV is from the *Rhabdoviridae* and is a negative-sense ssRNA virus, and SHTV is from the *Tetraviridae* and is a positive-sense ssRNA virus. To compare the small RNA responses generated by these distant ISVs, we deep-sequenced small RNAs from ovaries, testes and carcasses of *Ae. albopictus* triple-infected with AEFV, MERV and SHTV. All three viruses infected the reproductive tissues and carcasses. Each ISV showed a unique small RNA profile independent of the sex and tissue of the mosquito tested. Similar amounts of viral small RNAs were produced in the testes and carcasses, while the ovaries produced reduced amounts. We also observed differential gene expression for some PIWI-genes between reproductive tissues and the carcasses. Our results showed that *Ae. albopictus* responds to co-infecting ISVs in a virus-specific manner at the small RNA level and that male adult mosquitoes mount a similar small RNA response to that of females during viral infection. 

## 2. Materials and Methods

### 2.1. Cell Culture and Virus Production

C6/36 cells (ATCC CRL-1660) derived from *Ae. albopictus* were maintained at 28 °C in L-15 Leibovitz’s medium (Thermo Fisher Scientific, Waltham, MA, USA) supplemented with 10% fetal bovine serum (FBS; Thermo Fisher Scientific, Waltham, MA, USA), 1% nonessential amino acids (Gibco), 2% tryptose phosphate broth (Sigma, St. Louis, MO, USA), and 1% penicillin-streptomycin (Thermo Fisher Scientific, Waltham, MA, USA). The C6/36 cells used in this study are free of known viruses in the NCBI viruses database, confirmed by BLASTn searches using high throughput DNA and RNA sequencing data from the cells with the following parameters: reward/penalty value of 1/-1, gap opening penalty of 2, gap extension of 1, and the expected value of 0.001.

The mixed virus stock containing AEFV strain Narita-21, MERV isolate Menghai, and SHTV strain Shinobi was kindly provided by Haruhiko Isawa, National Institute of Infectious Diseases, Japan. The virus stock was amplified in naïve C6/36 cells grown in L-15 Leibovitz’s medium and the supernatant was used for the co-infection experiment in vivo. To create a standard curve for AEFV, the non-structural gene 4 (NS4) was amplified with primers containing the T7 promoter sequence followed by reverse transcription with MEGAscript T7 Transcription Kit (Thermo Fisher Scientific, Waltham, MA, USA) to generate AEFV NS4 RNA. The copy number of AEFV RNA was determined by reverse transcription-quantitative polymerase chain reaction (RT-qPCR) based on a serial dilution of the AEFV NS4 RNA with Maxima H minus First Strand cDNA Synthesis kit (Thermo Fisher Scientific, Waltham, MA, USA) and Fast SYBR Green Master Mix (Thermo Fisher Scientific, Waltham, MA, USA). MERV and SHTV RNA copy numbers in the virus stock were calculated using the Qgene template program [45].

### 2.2. Mosquitoes and Virus Infections

Laboratory colonies of *Ae. albopictus* were established using field-collected mosquitoes from Binh Duong Province, Vietnam (2011) [46] and Kawasaki, Kanagawa Prefecture Japan (2008) [47]. All the experiments were performed within 16–20 and 45–50 generations of laboratory colonization for the Vietnamese and Japanese mosquito colonies, respectively. The insectary conditions for maintenance of mosquitoes were 28°C, 70% relative humidity and a 12 h light: 12 h dark cycle. Adult mosquitoes were maintained with access to a 10% sucrose solution at all times. 

Four to seven days after emergence, adult female and male *Ae. albopictus* were intrathoracically injected with an AEFV/MERV/SHTV mixture containing 1 × 10^7^ genome copies of AEFV, 3.2 × 10^5^ genome copies of MERV, and 1.6 × 10^5^ genome copies of SHTV with a nanoinjector (Nanoject III, Drummond Scientific, Broomall, PA, USA). Following the injection, mosquitoes were incubated at 28 °C, 70% relative humidity and a 12 h light: 12 h dark cycle with access to a 10% sucrose solution at all times.

### 2.3. Quantification of ISVs, siRNA and piRNA Pathway-Related Gene RNA Levels in Co-infected Ae. albopictus Mosquitoes

Ovaries, testes and the rest of the bodies, which we define as carcasses for this study, from the AEFV/MERV/SHTV-injected mosquitoes were manually dissected at day 8 post-infection. Total RNA was extracted from 3 pools of 20 ovaries, testes or carcasses with TRIzol reagent (Thermo Fisher Scientific, Waltham, MA, USA). cDNA synthesis and qPCR were performed as described above for AEFV. To normalize ISV, DCR2, Ago2, and PIWI gene expression, two different housekeeping genes were used, actin and ribosomal protein L18 (RPL18). All PCR primer sequences are listed in Appendix A.

### 2.4. Small RNA Library Preparation

Three pools of 20 ovaries, testes or carcasses from the ISV-infected mosquitoes were combined for the generation of small RNA libraries. 19-33 nt long RNAs were cut and extracted from a 15% acrylamide/bis-acrylamide (37.5:1), 7M urea gel and the purified RNAs were subjected to small RNA library preparation using the NEBNext Multiplex Small RNA Library Prep (New England Biolabs, Ipswich, MA, USA) kit with a 3’ adaptor from Integrated DNA Technologies (Coralville, IA, USA) and in-house designed indexed primers. Libraries diluted to 4 nM were sequenced with the NextSeq 500 High Output Kit v2 (75 cycles) using a NextSeq 500 (Illumina, San Diego, CA, USA). Small RNA libraries have been submitted to the NCBI sequence read archive (SRA) under BioProject PRJNA587399.

### 2.5. Bioinformatics Analysis

The quality of the fastq files was assessed with FastQC software (www.bioinformatics.babraham.ac.uk/projects/fastqc/). Low-quality bases and adaptors were trimmed from each read using the Cutadapt program and only reads showing an acceptable quality (Phred score, 20) were retained. A second set of graphics was generated by the FastQC software using the trimmed fastq files. Reads were mapped to the genome sequences of AEFV strain Narita-21 (GenBank accession no. AB488408.1), MERV isolate Menghai (GenBank accession no. KX785335.1) and SHTV strain shinobi (GenBank accession no. LC270813.1) using Bowtie [48]. The default parameters for small RNAs were used, with the exception that one mismatch was allowed between each reading and the target (the −v 1 parameter). Bowtie generates results in the Sequence Alignment/Map (SAM) format. All SAM files were analyzed using the SAMtools package to produce bam indexed files. Homemade R scripts with Rsamtools and Shortreads in Bioconductor software were used for the analysis of the bam files to create graphs. The number of small RNA reads mapped each virus was normalized by the total number of reads in each small RNA library. To survey MERV-like DNA sequences in the genome of the *Ae. albopictus* Vietnam strain, a BLASTn search against its genomic DNA library, deposited in SRA under BioProject accession number PRJNA587399, was performed. 

## 3. Results

### 3.1. AEFV, MERV and SHTV Infection in Co-Injected *Ae. albopictus* Mosquitoes

*Ae. albopictus*-Vietnam (a colony derived from a field collection in Vietnam) adult females and males (4–7 days old post-emergence) were intrathoracically injected with a naturally derived viral stock containing a mixture of three ISVs. The relative genome copies for each virus injected in the mosquitoes were as follows: 1 × 10^7^ genome copies of AEFV, 3.2 × 10^5^ genome copies of MERV, and 1.6 × 10^5^ genome copies of SHTV. At 8 days post-injection (dpi), ovaries, testes and carcasses, which refers to the rest of the body parts without ovaries or testes, were dissected and 3 pools of 20 tissues each were prepared to examine ISV infections. RNA levels of each ISV were quantified by RT-qPCR and normalized based on the levels of actin or RPL18 RNA (Figure 1A,B). ISV RNA levels were very similar regardless of whether normalization was performed with actin or RPL18. Both AEFV and MERV infected the reproductive tissues and carcasses. SHTV infection was also observed in at least one of the pools of each tissue type, although the amount of SHTV RNA was one half to four logs lower than the RNA levels observed for AEFV and MERV. SHTV RNA was not detected in two pools of testes. AEFV and MERV RNA levels were comparable between both female and male carcasses, but SHTV RNA was higher in female carcasses than male carcasses. These findings are in agreement with a previous report showing that these ISVs do not compete with each other during infection in the *Ae. albopictus*-derived C6/36 cell line [14]. 

### 3.2. Small RNA Profiles in AEFV, MERV and SHTV-Co-Infected *Ae. albopictus* Mosquitoes

In order to characterize small RNA responses to AEFV, MERV, and SHTV in the co-infected *Ae. albopictus*-Vietnam mosquitoes, we generated small RNA libraries from a pooled RNA sample of the 3 pools used for quantification of ISV RNA levels. The size distribution of the total small RNAs from both female and male carcasses showed peaks at 21 and 22 nt corresponding to siRNAs and miRNAs, respectively (Figure 2A, blue). Ovaries and testes showed enrichment for 27-30 nt small RNAs corresponding to piRNAs (Figure 2A, red). Small RNA reads were mapped to each virus genome sequence to determine the size distribution of virus-derived small RNAs. In both reproductive tissues and carcasses, vsiRNAs of 21 nt were detected for each of the three viruses (Figure 2B and Appendix A); this was more pronounced in the carcasses of female mosquitoes. Ovaries produced approximately 0.39%, 0.18%, and 0.11% of the total amount of vsiRNAs derived from AEFV, MERV, and SHTV, respectively. Testes showed higher proportions of vsiRNAs from all three ISVs; 39%, 17% and 63% of total vsiRNAs from AEFV, MERV, and SHTV, respectively, were detected in this tissue. A peak of 26-30 nt-long viral small RNAs, potentially representing vpiRNAs, was observed only for MERV (Appendix A). To compare putative vpiRNA production among the three ISVs, we calculated the proportion of putative vpiRNAs to vsiRNAs by determining the number of virus-derived 26-30 nt-long reads per 100 virus-derived 21 nt reads (i.e., vsiRNAs) (Figure 2C). The proportion of MERV-derived putative piRNAs to MERV-derived vsiRNAs was between 30-90% and was similar between sexes and the tissues tested. AEFV-derived putative piRNAs were detected at lower levels; the proportion of putative AEFV-derived piRNAs to AEFV-derived siRNAs was 1-30% (Figure 2C and Appendix A). However, unlike putative MERV-derived vpiRNAs, which were produced from both strands, AEFV-piRNA-like small RNAs were predominantly generated from the positive-strand, i.e., from the strand corresponding to the viral genome. SHTV showed almost no production of 26-30 nt-long small RNAs, with a maximum of 2.5% compared to the vsiRNAs. To check that the absence of 26-30 nt-long small RNAs from SHTV was not due to the lower level of viral replication (Figure 1 and Figure 2B), we generated a new small RNA library from *Ae. albopictus*-Japan (a colony derived from a field collection in Japan) co-infected with the three ISVs by intrathoracic injection. Analysis of this library confirmed that, despite the high number of reads corresponding to SHTV-derived siRNAs, 26-32 nt-long small RNAs derived from SHTV were nearly undetectable in carcasses of female mosquitoes (Appendix A).

Overall, our results indicated that the type of virus-derived small RNAs produced during virus infection was virus-dependent but not sex- or tissue-dependent, however, the amounts of virus-derived small RNAs varied depending on the sex and on whether or not the infected tissue was a reproductive organ. 

### 3.3. Analysis of AEFV-Derived Small RNAs in the Co-Infected *Ae. albopictus*-Vietnam

To further characterize the small RNA profiles during infection with AEFV (a positive-sense ssRNA virus), the 21 nt vsiRNAs and 26-30 nt vpiRNA-like small RNAs were mapped to the AEFV genome sequence. The vsiRNAs were detected across the entire viral genome in carcasses with a notably high density of vsiRNAs mapping to the 3’ UTR (Figure 3A). The 26-30 nt small RNAs mostly mapped to the positive strand of AEFV, however, 26-30 nt small RNAs derived from the 3’ UTR mapped to both strands (Figure 3B). The detected number of 26-30 nt reads in ovaries was close to background levels. In contrast, a considerable number of 26-30 nt reads were observed in male and female carcasses and in testes. These 26-30 nt reads were primarily derived from the positive strand (Figure 2C and Figure 3B) and did not display the 1U or 10A biases characteristic of piRNAs (positions indicated by black or red arrowheads in the heat map, respectively), suggesting that they are not canonical vpiRNAs but most likely the result of degradation of viral RNA.

### 3.4. Analysis of SHTV -Derived Small RNAs in the Co-Infected *Ae. albopictus*-Vietnam

Mapping of the small RNA reads to the genome of SHTV (a positive-sense ssRNA virus) showed that the vast majority were vsiRNAs. SHTV-derived vsiRNAs mapped across the entire genome and lacked any obvious hotspots (Figure 4A). Very few SHTV-derived vsiRNAs (Figure 4A) and no 26-30 nt-long small RNAs (data not shown) were detected in ovaries, suggesting that SHTV infection does not result in the production of small RNAs in the female germline, probably due to a lack of replication in this tissue (Figure 1). With regard to putative vpiRNA production, 26-30 nt-long small RNA reads were observed in male and female carcasses and in testes with a random distribution throughout the genome (Figure 4B). No 1U or 10A biases were detected, suggesting that as for AEFV, these small RNAs are not vpiRNAs but are instead random products of viral RNA degradation.

### 3.5. Analysis of MERV-Derived Small RNAs in the Co-Infected *Ae. albopictus*-Vietnam

The small RNA profile of MERV (a negative-sense ssRNA virus) showed high levels of both vsiRNAs and putative vpiRNAs (Figure 2B,C and Appendix A). The 21-nt vsiRNAs were distributed across the viral genome with a hotspot of vsiRNAs derived from the negative strand in the 5’ UTR (Figure 5A). In contrast, the 26-30-nt small RNAs mostly mapped to structural genes (N, P, M, and G) within the MERV genome and were predominantly derived from the positive strand (Figure 5B). We observed a 1U bias for 26-30 nt reads derived from the positive strand (Figure 5B, black arrowheads) and 10A bias for 26-30 nt reads derived from the negative strand in all conditions (Figure 5B, red arrowheads), strongly suggesting that mosquitoes produced canonical MERV-vpiRNAs and induced ping-pong amplification in both reproductive tissues and carcasses. It has been reported that endogenous viral elements generate piRNAs in mosquitoes [49]. To examine whether MERV-derived piRNAs were a specific product of virus replication or derived from mosquito genomic DNA, we performed a BLASTn search for MERV-like DNA sequences in the *Ae. albopictus* genome (Bioproject PRJNA587399). No MERV-like DNA sequences were found in the *Ae. albopictus*-Vietnam genome. These results indicate that the vpiRNAs were derived from MERV RNA.

### 3.6. Expression of RNAi and PIWI Genes in Ae. albopictus Reproductive Organs and Carcasses

A previous transcriptomic study in *Ae. aegypti* found differential expression of some PIWI genes between ovaries and carcasses [50]. So far, no study has analyzed or reported the function of any of the PIWI genes in *Ae. albopictus* in vivo. This prompted us to determine if a link between the production of vpiRNAs and PIWI gene expression levels in co-infected mosquitoes existed. We examined mRNA levels of the main components of the piRNA pathway, together with the siRNA pathway, in ovaries, testes, and carcasses. Based on the *Ae. aegypti* PIWI genes, a recent bioinformatics study [35] indicated the existence of 10 members of the PIWI family in *Ae. albopictus* mosquitoes - Piwi1-9 and Ago3. Using the same RNA samples we used to measure ISV RNA levels in different mosquito tissues, we performed RT-qPCR to determine the expression levels of genes predicted to be involved in the production of siRNAs and piRNAs in *Ae. albopictus*. Due to the sequence similarity of some PIWI genes, we assessed gene expression as a group or as pairs for Piwi1-4, Piwi5/6, and Piwi8/9. Relative gene expression was calculated by normalization based on the levels of actin or RPL18 RNA. Overall, ovaries displayed a significantly higher level of expression for all genes examined, especially PIWI genes, as compared to other samples (Figure 6A,B), thus implying that both small RNA pathways are present and functional in ovaries. The most remarkable difference in relative gene expression levels between reproductive tissues and carcasses was observed for the Piwi1-4 group. We observed that Piwi1-4 group was almost exclusively expressed in ovaries and testes, but not in carcasses where vpiRNAs were largely produced. Although nothing is known about a direct correlation between the expression levels of Piwi1-4 and vpiRNA production, this observation strongly suggests that Piwi1-4 are not involved in vpiRNA production in the somatic tissues of *Ae. albopictus* females or males. 

## 4. Discussion

Recent advances in metagenomics have revealed that mosquitoes are abundantly infected with diverse ISVs in nature. However, studies of mosquito-virus interactions have mainly focused on arboviruses. To better understand the small RNA responses to ISVs in mosquitoes, we used *Ae. albopictus* mosquitoes co-infected with three distinct ISVs: AEFV, MERV, and SHTV. All three ISVs infect both female and male mosquitoes and are maintained in the population by vertical transmission. Taking advantage of this characteristic of ISVs, we examined whether adult male mosquitoes mount the same small RNA response as adult females in somatic and reproductive tissues where the piRNA pathway is known to be active. 

In the co-infected mosquitoes, RNA levels of all three ISVs were higher in the carcasses compared to the reproductive tissues. SHTV RNA levels varied among biological replicates and SHTV RNA levels were significantly lower than those of AEFV and MERV, even though the number of viral RNA copies injected into the mosquitoes was similar for all three viruses. Significantly larger amounts of SHTV-derived vsiRNAs and higher levels of SHTV viral RNA were observed in *Ae. albopictus*-Japan compared to *Ae. albopictus*-Vietnam. The reduced amount of SHTV RNA in the Vietnam colony could be due to differences in virus susceptibility between the mosquito strains. 

The overall small RNA profiles observed in ovaries and testes were visibly different from the profiles observed in carcasses and were characterized by a predominance of piRNAs compared to siRNAs and miRNAs. This is consistent with a previous study in *Ae. aegypti* [50]. Interestingly, testes exhibit a higher proportion of total and viral siRNAs to piRNAs when compared to ovaries. The number of siRNA reads derived from all three ISVs was significantly higher in the testes than the ovaries. This result suggests that siRNA production might be more active in the testes than in other tissues or that the testes have a greater capacity to internalize small RNAs from other virus-infected tissues and/or cells. 

The production of different types of small RNAs is in agreement with previous studies in *Aedes* or *Culex* mosquito cell lines persistently infected with ISVs, which showed the production of ISV-derived siRNAs and/or piRNAs [28,30,32]. While vsiRNA production was observed for all three ISVs, vpiRNAs with a clear ping-pong signature were only derived from MERV and these were found in both reproductive tissues and carcasses. Previous studies in *Aedes* and *Culex* mosquito cell lines [28,30,51,52] showed the production of vpiRNAs with ping-pong signatures during infection with the negative-sense single-stranded RNA viruses Phasi Charoen-like virus (*Phenuiviriade*), Merida virus (*Rhabdoviridae*), La Crosse virus, or Rift Valley fever virus (both *Bunyaviridae*). In addition, both primary and secondary vpiRNAs were generated from some alphaviruses (positive-sense single-stranded viruses) such as SINV, SFV, and chikungunya virus in vitro and/or in vivo in *Aedes* mosquitoes [15,26,27,51,53]. These observations suggest that small RNA responses to ISVs are highly virus-dependent, possibly related to their replication and transcription mechanisms, but independent of mosquito species or tissues. 

The vsiRNAs derived from AEFV and MERV clustered in the 3’ UTR and 5’ UTR, respectively. This accumulation was also observed for 26-30 nt small RNAs, although the number of reads was much lower than the number of vsiRNAs. We speculate that these viral small RNAs from hotspots could derive from the secondary structure within the viral genome rather than from replication intermediates. DCR2 could cleave the hairpin structure of viral RNAs and the loop structure adjacent to the cleavage site might result in 26-30 nt long small RNAs as a by-product of vsiRNA production. Interestingly, previous studies suggested that the subgenomic RNA of flaviviruses, which corresponds to the 3’ UTR, is capable of suppressing the RNAi machinery, possibly by acting as an RNA decoy or by inhibiting the loading of vsiRNA into the RISC [21,23,24]. The high amount of siRNAs from the 3’ UTR of AEFV could be a strategy for the virus to escape from the siRNA pathway. Further analyses are required to explore this hypothesis.

MERV-derived piRNAs were largely produced from the region of the viral genome encoding the structural proteins, corresponding to nucleotide positions 130 to 4162. *Cx. quinquefasciatus* cell lines infected with Merida virus, also a rhabdovirus, showed a similar vpiRNA profile, suggesting that the regions that generate vpiRNAs are conserved regardless of the mosquito species and depend on the viral replication strategy [30]. Rhabdoviruses, in general, transcribe their mRNA from the 3’ UTR, which is closer to the structural genes. The structural gene transcripts might thus be more efficiently targeted by the piRNA pathway than the non-structural gene transcript transcribed from the 5’ end. Interestingly, different studies showed that vpiRNAs from alphaviruses, orthobunyaviruses, and phelobovirus were also largely generated from viral genomic regions encoding structural genes in vitro and/or in *Aedes* mosquitoes [15,26,27,51,52,53]. It remains to be elucidated how exogenous viral RNAs are recognized as precursors for vpiRNA biogenesis, however, the transcription mechanism of viral structural genes might be involved in recognition by PIWI proteins. In *Ae. aegypti*, Piwi5 is likely to play a role in producing primary piRNAs from both TEs and exogenous viruses, however, how Piwi5 selects the transcript to process into primary piRNAs is still unknown [26]. 

A recent study identified ten PIWI genes, Piwi1-9 and Ago3, in *Ae. albopictus* based on PIWI genes in *Ae. aegypti* using a tBLASTx search [35]. However, it is important to note that the genome assembly and gene annotation of *Ae. albopictus* is not as well developed as compared to *Ae. aegypti*. We quantified the expression levels of the putative Piwi1-9 and Ago3 genes as well as the DCR2 and Ago2 genes by qPCR in ISVs-infected *Ae. albopictus* samples. A previous study examined PIWI gene expression in midgut and non-midgut tissues of female *Ae. albopictus* mosquitoes by transcriptome analysis and RT-PCR [35]. Our results were consistent with this study, for instance, Piwi1-4 were abundantly expressed in the reproductive tissues, which correspond to the non-midgut tissue samples of the transcriptome analysis. Our qPCR analysis detected Piwi8/9 expression in both adult reproductive tissues and the carcasses, while their expression was only observed in embryos and larvae stage by RT-PCR in the previous study. This difference could be explained by the sensitivity of the assay. Interestingly, our study showed that Piwi1-4 were exclusively expressed in the reproductive tissues, which showed significantly less MERV-derived vpiRNAs than the carcass. This observation indicates that Piwi1-4 do not play a role in vpiRNA production. Instead, vpiRNA biogenesis must require one or more of the of Piwi5-9 proteins in *Ae. albopictus*. Nevertheless, a better genome assembly and annotation for *Ae. albopictus* is required to conduct studies that will allow the determination of the PIWI genes involved in vpiRNA biogenesis.

The present study demonstrated that, regardless of the mosquito sex and the reproductive nature of the tissue, the production of different types of viral small RNAs from ISVs is virus-dependent. Our results highlight the value of ISVs as models to better understand small RNA pathways in mosquitoes as well as the potential countermeasures of viruses to escape the RNA interference response. Future studies should address the small RNA responses to ISVs in mosquitoes that are naturally infected. This could lead to new insights into mosquito immunity and mosquito–ISV–arbovirus relationships. 

## Figures and Tables

**Figure 1 viruses-12-00468-f001:**
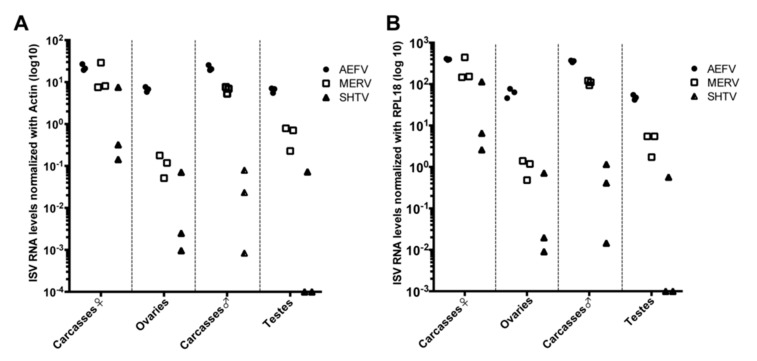
AEFV, MERV and SHTV infections in reproductive tissues (ovaries and testes) and carcasses. *Ae. albopictus* adult females and males were intrathoracically injected with an AEFV/MERV/SHTV mixture and collected at 8 days post-injection. RNA levels for each ISV in the co-infected mosquito ovaries, testes or carcasses were quantified by RT-qPCR normalized with actin (**A**) or RPL18 RNA (**B**). Each dot represents a pool of 20 reproductive tissues or carcasses.

**Figure 2 viruses-12-00468-f002:**
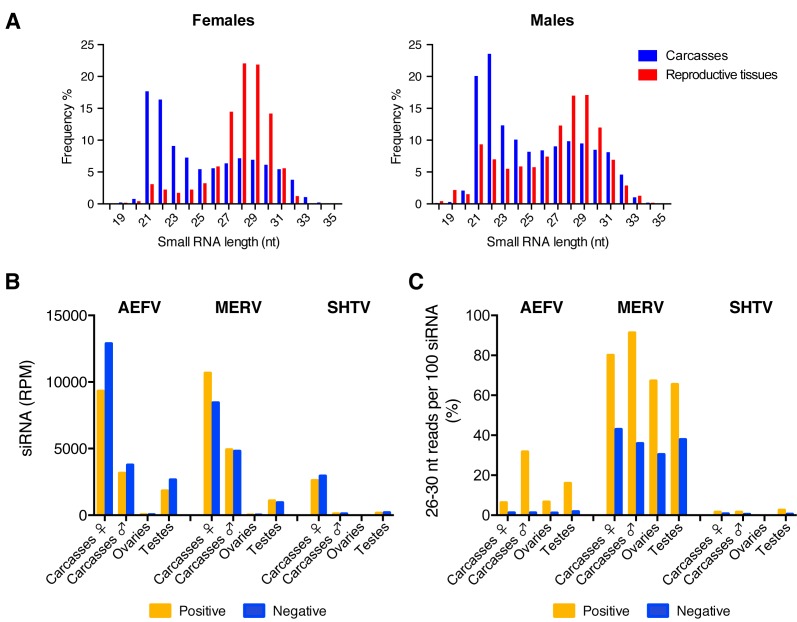
Comparison of vsiRNA and putative vpiRNA production from ISVs in reproductive tissues and carcasses of female or male *Ae. albopictus* mosquitoes. The size distribution of total small RNA reads from each sample is shown in (**A**). Normalized siRNA reads per one million (RPM) mapped reads to the AEFV, MERV or SHTV genome are shown in (**B**). Proportion of 26-30 nt AEFV-, MERV- or SHTV-derived small RNAs per 100 vsiRNAs is represented in (**C**). Yellow and blue bars represent positive- and negative-stranded reads, respectively.

**Figure 3 viruses-12-00468-f003:**
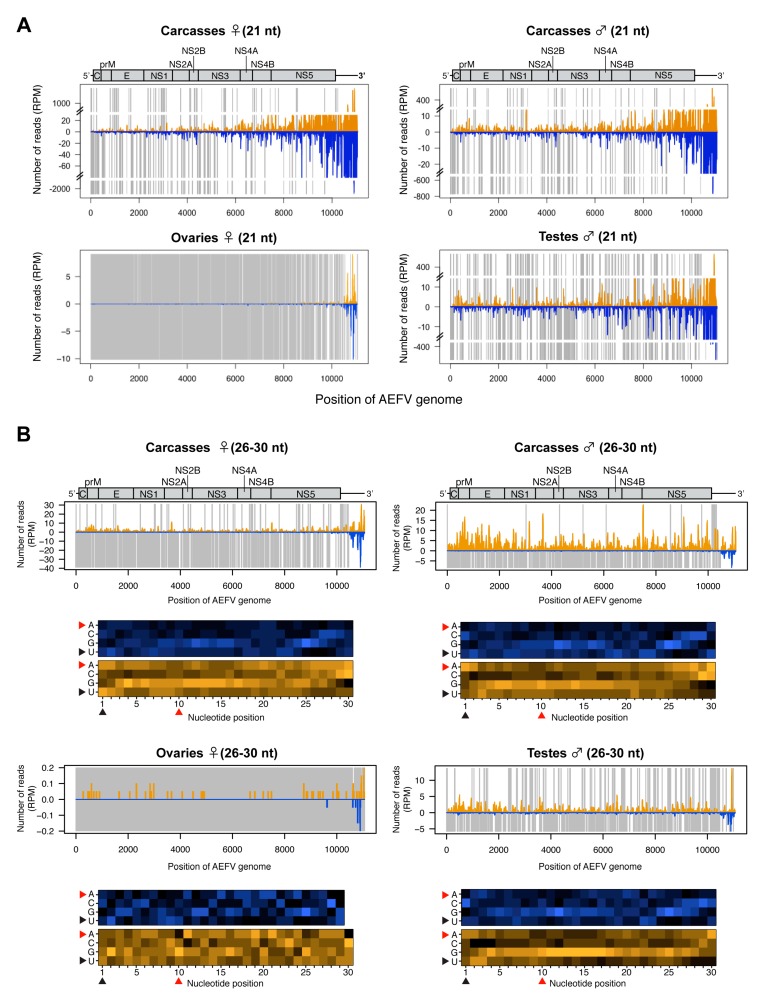
Small RNA responses to AEFV in female and male *Ae. albopictus* reproductive tissues and carcasses. The distribution of siRNA (21 nt) (**A**) and piRNA-like small RNAs (26-30 nt) (upper panel, **B**) mapped to the AEFV genome. A schematic illustration of the AEFV genome is shown on top of small RNAs mapping to the viral sequence. Yellow and blue bars represent positive- and negative-stranded reads, respectively. Regions with no coverage are indicated by gray bars. For AEFV-derived 26-30 nt piRNA-like small RNAs, relative nucleotide frequency at each position is shown as a heat map (lower panel in B) in which the color intensity denotes the frequency. The black and red arrowheads point to the 1U and 10A positions, respectively.

**Figure 4 viruses-12-00468-f004:**
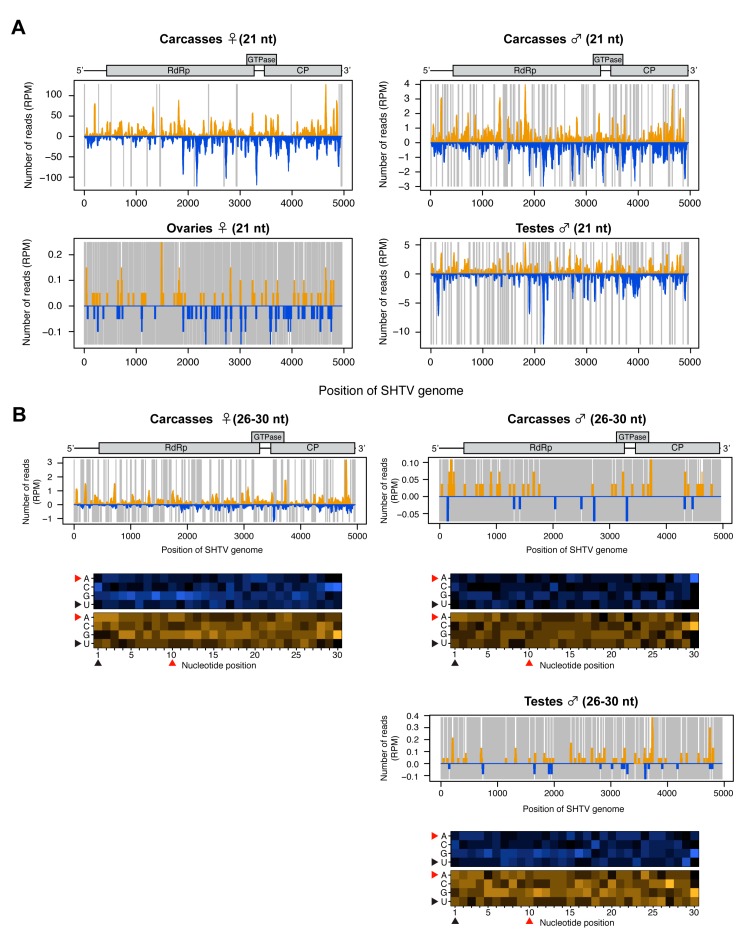
Small RNA responses to SHTV in female and male *Ae. albopictus* reproductive tissues and carcasses. The distribution of siRNA (21 nt) (**A**) and piRNA-like small RNAs (26-30 nt) (upper panel, **B**) mapped to the SHTV genome. A schematic illustration of the SHTV genome is shown on top of small RNAs mapping to the viral sequence. Yellow and blue bars represent positive- and negative-stranded reads, respectively. Regions with no coverage are indicated by gray lines. For 26-30 nt SHTV-derived piRNA-like small RNAs, relative nucleotide frequency at each position is shown as a heat map (lower panel in **B**) in which the color intensity denotes the frequency. 26-30 nt small RNA reads were not detected in ovary samples (**B**). The black and red arrowheads point to the 1U and 10A positions, respectively.

**Figure 5 viruses-12-00468-f005:**
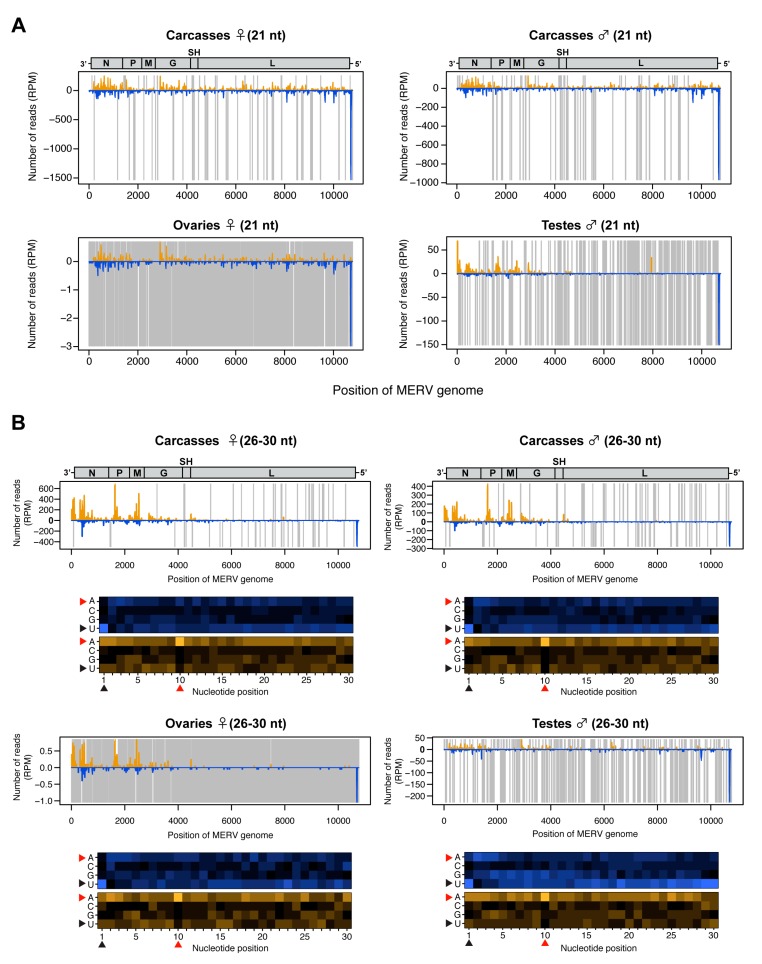
Small RNA responses to MERV in female and male *Ae. albopictus* reproductive tissues and carcasses. The distribution of siRNA (21 nt) (**A**) and piRNA-like small RNAs (26-30 nt) (upper panel, **B**) mapped to the MERV genome. A schematic illustration of the MERV genome is shown on top of small RNAs mapping to the viral sequence. Yellow and blue bars represent positive- and negative-stranded reads, respectively. Regions with no coverage are indicated by gray lines. For 26-30 nt MERV-derived piRNA-like small RNAs, relative nucleotide frequency at each position is shown as a heat map (lower panel in **B**), in which the color intensity denotes the frequency. The black and red arrowheads point to the 1U and 10A positions, respectively.

**Figure 6 viruses-12-00468-f006:**
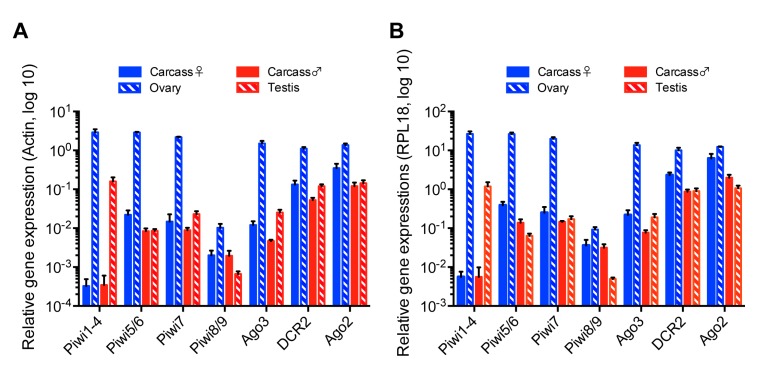
Relative gene expression of siRNA and piRNA pathway components in reproductive tissues and carcasses of *Ae. albopictus* females and males co-infected with AEFV, MERV, and SHTV. Relative gene expression of the PIWI genes Piwi1-9 and Ago3, together with DCR2 and Ago2, key components of the siRNA pathway, was examined by RT-qPCR. Relative gene expression was calculated by normalization with actin (**A**) or RPL18 RNA (**B**). Multiple unpaired t-test with Holm-Sidak correction was applied to calculate the statistical significance. The P values are summarized in Appendix A.

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
