# Peer review of "Differential Small RNA Responses against Co-Infecting Insect-Specific Viruses in Aedes albopictus Mosquitoes"

_viruses, 2020, doi:10.3390/v12040468_

Round 1

Reviewer 1 Report

Small RNAs produced during arbovirus infection of mosquitoes have been characterised in numerous studies in female mosquitoes. In the current study, Frangeul and colleagues use insect-specific viruses as an interesting tool to compare the small RNAs produced in male and female Aedes albopictus mosquitoes during infection with three insect-specific viruses. The study provides new insights into the RNAi response to insect-specific viruses, and also provides new information on how the RNAi response compares in male and female mosquitoes, with a specific emphasis on the reproductive organs. Overall, I find that the manuscript is clearly written, the data well-presented and the conclusions justified.

I have only minor comments on the work:

  • I thought that in certain points the materials and methods lacked the level of detail necessary to fully understand and replicate the work. Specifically:
    1. Line 104: what parameters were used in BLAST searches to define C6/36 as being virus-free?
    2. Line 117: how and when were the mosquito colonies generated? How many generations have the colonies gone through in a laboratory environment? Is there a paper with more information that the authors could cite? This is important because these colonies are not standard established lines commonly used in the field (which is also a strength of the study).
    3. Line 153: can the home-made R scripts be made public? The analysis will be difficult to reproduce if more details on scripts are not provided.
    4. Line 161 (Results): at how many days post-emergence were adult mosquitoes infected? Immune responses might be expected to differ in ‘young’ versus ‘old’ mosquitoes and this information would help compare data to other studies.
    5. Line 206 (Results): were the infection methods for the Japanese-derived mosquito colony the same as for the Vietnamese-derived colony? More detail would be helpful here.
  • For data presented in Fig 1, is there a way of demonstrating that the detected viral RNA is really due to viral replication and not simply input virus? The infectious titre was specified, but how this relates to the normalised titres 8 days post-infection is not clear to me.
  • Line 200 (Fig 2C): it seems to me that there is also a piRNA bias towards the positive-strand in MERV.
  • In the heatmaps showing 1U/10A bias (or not), should the arrow in the lower panel (yellow) be pointing to the U or the A? Seems like it should point to the A, but perhaps I misunderstand the emphasis intended. If this is an error, it is consistent in all the figures.
  • Fig 6: I think it would be more intuitive and helpful to incorporate the statistics in the figure rather than specifying in the supplementary material.

Author Response

Response to Reviewer 1 Comments
Comment: Small RNAs produced during arbovirus infection of mosquitoes have been characterised in numerous studies in female mosquitoes. In the current study, Frangeul and colleagues use insect-specific viruses as an interesting tool to compare the small RNAs
produced in male and female Aedes albopictus mosquitoes during infection with three insect-specific viruses. The study provides new insights into the RNAi response to insectspecific viruses, and also provides new information on how the RNAi response compares in
male and female mosquitoes, with a specific emphasis on the reproductive organs. Overall, I find that the manuscript is clearly written, the data well-presented and the conclusions justified.

I have only minor comments on the work:
Answer: We thank the reviewer for the thoughtful comments. We have taken into account the reviewer’s suggestions to strengthen the manuscript. We have addressed the reviewer’s specific concerns as follows:

Comment: I thought that in certain points the materials and methods lacked the level of detail necessary to fully understand and replicate the work. Specifically:
1. Line 104: what parameters were used in BLAST searches to define C6/36 as being virusfree?
Answer: To define our C6/36 cell lines as being “virus-free” from known viruses, we used highthroughput DNA and RNA sequencing data from the cell line available in the lab. Assemblies of these data were prepared (using Spades 3.8) and the similarity of the assembled contigs to viral sequences was determined with a non-stringent BLASTn search, using a matrix of 1/-1 with a gap opening penalty of 2, a gap extension of 1 and an expected value of 0.001. These
options can identify any contig that has at least 60% similarity to a viral sequence. We have now added the parameters for BLASTn search in lines 103 - 106 of the main text.

Comment:
2. Line 117: how and when were the mosquito colonies generated? How many generations have the colonies gone through in a laboratory environment? Is there a paper with more information that the authors could cite? This is important because these colonies are not standard established lines commonly used in the field (which is also a strength of the study).

Answer: Regarding the Vietnamese Ae. albopictus colony, we added the generation of the colony, the specific location and the reference mentioning that the eggs were collected by ovitraps. Unfortunately, we do not have an exact record for the number of generations for the Japanese Ae. albopictus colony, as our collaborator did not provide us with this information. However, we retrospectively calculated the approximate number of generations as 45-50 based on the maintenance frequency of this mosquito colony. We added this
information and the specific location for collection in the main text. We also refer to a paper that used this Japanese Ae. albopictus colony, although there is no detail about how the colony was generated. This new information is found on lines 119 – 122 in the main text.

Comment:
3. Line 153: can the home-made R scripts be made public? The analysis will be difficult to reproduce if more details on scripts are not provided.

Answer: The home-made R scripts are specifically used for visualization of the data (i.e. histograms and heatmaps), but not for the analysis itself. All the analysis was carried out with “Samtools,” which is a commonly used program. We believe that it is less useful to publicly share our R scripts, as there are already some freely distributed scripts, such as Virome (PMID:23709497), which produce the same types of graphs.

Comment:
4. Line 161 (Results): at how many days post-emergence were adult mosquitoes infected? Immune responses might be expected to differ in ‘young’ versus ‘old’ mosquitoes and this information would help compare data to other studies.

Answer: The adult mosquitoes were used for the experiments at 4-7 days after emergence. This information was originally presented in the Materials and Methods section. To make it clearer, we modified the sentences on line 125 in Materials and Methods section, and on line
166 in the Result section.

Comment:
5. Line 206 (Results): were the infection methods for the Japanese-derived mosquito colony the same as for the Vietnamese-derived colony? More detail would be helpful here.

Answer: We thank the reviewer for this comment. Yes, the infection method was exactly the same for Japanese and Vietnamese Ae. albopictus (infection by intrathoracic injection). We added this information on line 213.

Comment: For data presented in Fig 1, is there a way of demonstrating that the detected viral RNA is really due to viral replication and not simply input virus? The infectious titre
was specified, but how this relates to the normalised titres 8 days post-infection is not clear to me.

Answer: The detection of a large number of viral small RNAs, especially siRNAs, from the three ISVs tested in this study indicates that these ISVs replicate well in the mosquito. In Figure 1, we focus on the comparison of 1) viral RNA levels among the three different ISVs
and 2) ISV RNA levels among the different mosquito samples. For these purposes, we thought relative quantification was more appropriate than calculation of absolute viral RNA copy numbers. We used two different housekeeping genes to avoid misinterpreting viral RNA levels in each mosquito sample due to potential differences in the expression of the housekeeping gene. Such differences were not observed in this study.

Reviewer 2 Report

Several recent studies suggest important role for insect-specific viruses in the interactions with arboviruses transmitted by mosquitoes. This study did a nice job in examining the in vivo small RNA responses of mosquitoes to three distinct insect-specific viruses, providing a foundation for future studies. The only suggestion I would like to make is for the authors to use the total mapped reads (i.e., all reads mapped to the genomes of the mosquito and viruses) to normalize the relative abundance of the vsiRNAs and vpiRNAs from the three viruses.

Author Response

Response to Reviewer 2 Comments
Comment: Several recent studies suggest important role for insect-specific viruses in the interactions with arboviruses transmitted by mosquitoes. This study did a nice job in examining the in vivo small RNA responses of mosquitoes to three distinct insect-specific viruses, providing a foundation for future studies. The only suggestion I would like to make is for the authors to use the total mapped reads (i.e., all reads mapped to the genomes of the mosquito and viruses) to normalize the relative abundance of the vsiRNAs and vpiRNAs from the three viruses.

Answer: We thank the reviewer for his/her suggestions. We have carefully considered the reviewer’s suggestion to normalize the relative abundance of vsiRNAs and vpiRNAs using total mapped reads. However, we still believe that the current normalization is more appropriate to our study for the following reasons:
In the current state of bioinformatics and next generation sequencing technology, there is not an absolutely correct way to normalize small RNA reads for comparisons among data sets. The appropriate approach needs to be selected for each aim. In our study, we focused on a qualitative and quantitative comparison of the small RNA responses to three different viruses in the reproductive tissues and carcasses of coinfected mosquitoes. The replication rate of each virus and the efficiency with which each virus is targeted by RNA interference is different for each virus and varies among different tissue types. Normalization based on total mapped reads obscures important components of the small RNA response. For example, if we normalize the data shown in Figure 2C using total mapped reads, the production of MERVderived piRNAs in ovaries and testes is not readily observed. This apparent lack of MERVderived piRNAs in the reproductive tissues does not indicate the absence of a piRNA-based response to MERV in these tissues, but rather is a reflection of the low replication rate of ERV in ovaries and testes. Normalization based on vsiRNAs accounts for the low level of MERV RNA in reproductive tissues and facilitates the observation that canonical vpiRNAs were generated from MERV, but not from the other two viruses independent of the tissues tested. Thus, we believe that normalizing with vsiRNAs derived from each virus is the most appropriate approach. We hope this explanation addresses the reviewer’s concern.